# *Limosilactobacillus reuteri* HY7503 and Its Cellular Proteins Alleviate Endothelial Dysfunction by Increasing Nitric Oxide Production and Regulating Cell Adhesion Molecule Levels

**DOI:** 10.3390/ijms252011326

**Published:** 2024-10-21

**Authors:** Hyejin Jeon, Daehyeop Lee, Joo-Yun Kim, Jae-Jung Shim, Jae-Hwan Lee

**Affiliations:** R&BD Center, Hy Co., Ltd., 22 Giheungdanji-ro 24 Beon-gil, Giheung-gu, Yongin-si 17086, Gyeonggi-do, Republic of Korea; 10003012@hy.co.kr (H.J.); flywhy7@hy.co.kr (D.L.); jjshim@hy.co.kr (J.-J.S.); jaehwan@hy.co.kr (J.-H.L.)

**Keywords:** angiotensin II, endothelial dysfunction, lactic acid bacteria, *Limisilactobacillus reuteri*, probiotics

## Abstract

Endothelial dysfunction, which is marked by a reduction in nitric oxide (NO) production or an imbalance in relaxing and contracting factor levels, exacerbates atherosclerosis by promoting the production of cell adhesion molecules and cytokines. This study aimed to investigate the effects of *Limosilactobacillus reuteri* HY7503, a novel probiotic isolated from raw milk, on endothelial dysfunction. Five lactic acid bacterial strains were screened for their antioxidant, anti-inflammatory, and endothelium-protective properties; *L. reuteri* HY7503 had the most potent effect. In a mouse model of angiotensin II-induced endothelial dysfunction, *L. reuteri* HY7503 reduced vascular thickening (19.78%), increased serum NO levels (226.70%), upregulated endothelial NO synthase (*eNOS*) expression in the aortic tissue, and decreased levels of cell adhesion molecules (intercellular adhesion molecule-1 [ICAM-1] and vascular cell adhesion molecule-1 [VCAM-1]) and serum cytokines (tumor necrosis factor-alpha [TNF-α] and interleukin-6 [IL-6]). In TNF-α-treated human umbilical vein endothelial cells (HUVECs), *L. reuteri* HY7503 enhanced NO production and reduced cell adhesion molecule levels. In HUVECs, surface-layer proteins (SLPs) were more effective than extracellular vesicles (exosomes) in increasing NO production and decreasing cell adhesion molecule levels. These findings suggested that *L. reuteri* HY7503 may serve as a functional probiotic that alleviates endothelial dysfunction.

## 1. Introduction

Endothelial cells, which form the inner lining of blood vessels, play a crucial role in maintaining vascular homeostasis by regulating nitric oxide (NO) production, a key factor in vasodilation, thrombosis prevention, and inflammation control, and NO is synthesized by endothelial NO synthase (eNOS), which is primarily expressed in endothelial cells [1]. Endothelial dysfunction, caused by oxidative stress and inflammatory mediators, leads to a decrease in eNOS activity and NO bioavailability and the disruption of NO production [2]. Endothelial dysfunction is characterized by the overexpression of adhesion molecules, including intercellular adhesion molecule-1 (ICAM-1) and vascular cell adhesion molecule-1 (VCAM-1) [3]. Another factor that causes endothelial dysfunction is angiotensin II (Ang II), a key peptide in the renin–angiotensin system. This causes hypertension, vascular hypertrophy, endothelial dysfunction, and eNOS dysfunction/coupling, which lowers NO levels and increases superoxide production; this affects vascular tension, inflammation, and procoagulant factor levels [4]. Therefore, endothelial dysfunction contributes to the progression of conditions such as hypertension, diabetes, inflammatory bowel disease, atherosclerosis, and other cardiovascular diseases (CVDs) [5]. Endothelial dysfunction has been utilized as a diagnostic and predictive tool for various diseases; it has also been widely used as a target during the early stages of treatment to slow disease progression [6].

Probiotics, especially lactic acid bacteria (LAB), have attracted attention not only for improving gut health but also for their potential broader health benefits, including their role in modulating inflammation, oxidative stress, and metabolic disorders [7,8]. Among these, *Lactobacillus* spp. (recently reclassified into multiple genera) have shown promising antioxidant and anti-inflammatory properties, making them attractive candidates for the prevention and management of chronic diseases [9]. These LAB strains are mainly present in fermented foods and dairy products, particularly in raw milk [10]. Recently, studies on LAB strains with specific functional properties have increased the interest in these strains as these can mitigate endothelial dysfunction [11]. Previous studies have demonstrated that LAB and yeast complexes restore the reactive oxygen species (ROS)/NO imbalance and improve endothelial dysfunction in hypertensive rats [11]. However, the specific mechanisms by which the individual strains and their bioactive substances exert these effects have not yet been studied.

Surface-layer proteins (SLPs), extracellular vesicles (exosomes), and exopolysaccharides are the bioactive substances in LAB strains that contribute to their biological effects [12,13,14]. SLPs are supramolecular cell envelope structures consisting of a single species of protein or multiple species of glycoproteins; previous studies have shown that SLPs have a variety of functions, including the inhibition of intestinal cell adhesion of pathogenic bacteria and the regulation of cytokine-related responses and bacteriolytic enzymes [15]. Exosomes are membrane structures that transport metabolic intermediates, proteins, and RNAs as cargo; they are produced by most LAB strains and have biological activities [16]. Their typical activities are similar to those of the bacteria that produce them, such as host immune cell stimulation and antibacterial functions [17,18]. Studies on the use of SLPs and exosomes as bioactive substances in relation to the functionality of LAB strains have increased in recent years; however, the mechanisms behind LAB effects on endothelial function remain unstudied. Therefore, in this study, we aimed to analyze the potential of LAB strains isolated from raw milk in terms of their vascular endothelium-protective activities and identify the bioactive substances present in these LAB strains. The LAB strain Limosilactobacillus reuteri HY7503, which has excellent antioxidant and anti-inflammatory effects and increases NO production in vascular endothelial cells, was selected for in vitro experiments. Furthermore, the effect of L. reuteri HY7503 in vivo was verified using mice with Ang II-induced endothelial dysfunction. In addition, the mechanisms of action and bioactive components of *L. reuteri* HY7503 were examined through in vitro experiments using human umbilical vein endothelial cells (HUVECs).

## 2. Results

### 2.1. Screening of Lactic Acid Bacterial Strains Isolated from Raw Milk

Five LAB strains were isolated from raw milk. Two of the strains belonged to *Lacticaseibacillus paracasei* and three belonged to *Limosilactobacillus reuteri*. The antioxidant activities of the five LAB strains were evaluated by measuring their 2,2-diphenyl-1-picrylhydrazyl (DPPH) radical scavenging activities. The DPPH radical scavenging activities of the isolated strains ranged from 4.31 to 21.96% (Figure 1) and increased in a dose-dependent manner. *L. reuteri* #1 showed the highest activity (21.96 ± 2.19%) compared to that shown by the other strains at the same dose (1 × 10^8^ colony-forming units [CFU]/mL).

The 3-(4,5-dimethylthiazol-2-yl)-2,5-diphenyltetrazolium bromide (MTT) assay was conducted using RAW264.7 cells. The selected strains were non-toxic at 1 × 10^5^ to 1 × 10^6^ CFU/mL, with a cell viability of approximately 98.63–122.06% (Figure 2A). Lipopolysaccharides (LPSs) were used to induce NO production due to inflammation in RAW264.7 cells; NO production was significantly enhanced to 216.63 ± 11.33% in these cells compared to that in the control (100 ± 3.72%; *p* < 0.001; Figure 2B). All selected strains significantly inhibited NO production compared to that observed in the LPS group. *L. paracasei* #4 exhibited the highest level of NO inhibition (77.25 ± 6.65%; *p* < 0.001), followed by that shown by *L. reuteri* #1 (92.68 ± 3.38%) and *L. reuteri* #2 (94.42 ± 4.85%; Figure 2B).

The results of the MTT assay, conducted using HUVECs, showed that the selected strains were non-toxic at concentrations of 1 × 10^5^ to 1 × 10^6^ CFU/mL, with cell viability values of approximately 96.43–192.86% (Figure 3A). NO production significantly decreased to 33.35 ± 19.23% in HUVECs treated with tumor necrosis factor-alpha (TNF-α) compared to that in the control (100 ± 9.57%; *p* < 0.001; Figure 3B). Of the selected strains, only two strains significantly enhanced NO production compared to that in the TNF-α-treated group. Cells treated with *L. reuteri* #1 exhibited the highest NO production rate (111.52 ± 24.69%; *p* < 0.001), followed by those treated with *L. reuteri* #2 (93.7 ± 8.98%; Figure 3B).

The strain with excellent antioxidant and anti-inflammatory activities that resulted in the maximum degree of enhancement of NO production in HUVECs was *L. reuteri* #1. Therefore, we used *L. reuteri* #1 for further experiments and named it HY7503.

### 2.2. L. reuteri HY7503 Decreases Aortic Wall Thickness in Mice with Angiotensin II-Induced Endothelial Dysfunction

To evaluate the effect of *L. reuteri* HY7503 on the vasculature in a mouse model, eight-week-old mice were fed two different doses (1 × 10^8^ and 1 × 10^9^ CFU/kg/day) of *L. reuteri* HY7503 or saline for five weeks. One week after the start of dietary intake, all groups, except the control (CON) group, were intraperitoneally injected daily with Ang II for four weeks to induce endothelial dysfunction. Weight loss was observed in the Ang II-treated group after week 1; a significant decrease in weight was observed after week 3 in the Ang II-treated group compared to that in the CON group (*p* < 0.05; Figure 4A). However, no significant differences in food intake were observed among the groups (Figure 4B).

Therefore, the thickness of the media (MT) and the ratio of MT to lumen diameter (LD) were determined via histological analysis of the aortic tissue (Figure 4C). The MT was the highest in the angiotensin II-treated group (Ang II; 45.9 ± 2.72 µm), and the thickness was significantly different compared to that observed in the CON group (36.37 ± 2.53 µm; *p* < 0.001; Figure 4D). On the other hand, the MT values for the groups of mice treated with 10^8^ and 10^9^ CFU/kg/day of *L. reuteri* HY7503 (HY7503-L and HY7503-H, respectively) decreased in a dose-dependent manner (39.78 ± 1.99 and 36.82 ± 2.01 µm, respectively), and the values were significantly different compared to that observed in the Ang II group (*p* < 0.001; Figure 4D). The MT/LD of the Ang II group was 7.78 ± 0.46%, which was higher than that of the CON group (2.09 ± 0.53%; *p* < 0.001; Figure 4E). Furthermore, the MT/LD values of the HY7503-L and HY7503-H groups were 6.78 ± 0.34 and 6.37 ± 0.35%, respectively, and these values were significantly lower than those of the Ang II group (*p* < 0.001; Figure 4E).

### 2.3. L. reuteri HY7503 Decreases NO Production and Downregulates Adhesion Molecule-Related Gene Expression in Mice with Angiotensin II-Induced Endothelial Dysfunction

To determine the effect of *L. reuteri* HY7503 on NO production during endothelial dysfunction, we analyzed NO levels in the serum and eNOS gene expression in the aortic tissue. Serum NO levels were confirmed using the Griess reagent assay. In the Ang II group, the NO level was the lowest at 41.67 ± 2.6%, which was significantly different from that in the CON group (100 ± 11.23%; *p* < 0.001; Figure 5A). NO levels in the HY7503-L and HY7503-H groups increased significantly (94.48 ± 5.15; *p* < 0.001 and 88.35 ± 18.45%; *p* < 0.05, respectively) compared to those in the Ang II group (*p* < 0.001 and *p* < 0.05, respectively; Figure 5A). Subsequently, the expression level of *eNOS* was examined using real-time polymerase chain reaction (RT-PCR) analysis. The *eNOS* mRNA level in the Ang II group was significantly lower than that in the CON group (*p* < 0.05; Figure 5B). However, the intake of *L. reuteri* HY7503 inhibited the Ang II-induced reduction in *eNOS* levels (*p* < 0.05; Figure 5B).

Endothelial dysfunction is associated with the overexpression of *ICAM-1* and *VCAM-1* [3]. The *ICAM-1* mRNA level in the Ang II group was significantly higher than that in the CON group (*p* < 0.05; Figure 6A). However, the intake of *L. reuteri* HY7503 resulted in a decrease in the *ICAM-1* levels (*p* < 0.05). In addition, the expression levels of *VCAM-1* increased significantly in the Ang II group compared to those in the CON group (*p* < 0.05; Figure 6B) and decreased significantly upon the ingestion of HY7503 (*p* < 0.005).

### 2.4. L. reuteri HY7503 Inhibits Inflammatory Cytokine Production in Mice with Angiotensin II-Induced Endothelial Dysfunction

Serum cytokine levels were determined using an enzyme-linked immunosorbent assay (ELISA). Serum levels of TNF-α were higher in the Ang II group compared to those in the CON group (*p* < 0.05) (Figure 7A). However, the ingestion of *L. reuteri* HY7503 led to a decrease in serum levels of TNF-α in a dose-dependent manner (*p* < 0.05) (Figure 7A). Similarly, IL-6 levels in the serum increased significantly in the ANG II group compared to those in the CON group (*p* < 0.001) (Figure 7B); however, the intake of *L. reuteri* HY7503 resulted in a significant dose-dependent decrease in serum IL-6 levels (*p* < 0.001) (Figure 7B).

### 2.5. L. reuteri HY7503 Upregulates Endothelial NOS-Related Gene Expression and Downregulates Adhesion Molecule-Related Gene Expression in Damage-Induced HUVECs

Based on the results of the animal experiments, we aimed to confirm the mechanism using a HUVEC model. Endothelial dysfunction was induced by treating HUVECs with TNF-α; subsequently, the cells were treated with various concentrations of *L. reuteri* HY7503 to examine changes in the gene expression associated with endothelial dysfunction. In the TNF-α-treated group, the *eNOS* level was significantly lower (*p* < 0.001; Figure 8A) and *ICAM-1* and *VCAM-1* levels were significantly higher (*p* < 0.001; Figure 8B,C) than those in the control group. In contrast, the groups treated with *L. reuteri* HY7503 exhibited an increase in *eNOS* expression levels (*p* < 0.05; Figure 8A) and a decrease in *ICAM-1* and *VCAM-1* expression levels (*p* < 0.01; Figure 8B,C) compared to those in the control group. Notably, the most significant changes were observed when the cells were treated with a 1 × 10^6^ CFU/mL concentration of *L. reuteri* HY7503 (*p* < 0.001). These results are consistent with the findings observed using aortic tissues in animal experiments.

### 2.6. Effects of Surface-Layer Proteins and Extracellular Vesicles of L. reuteri HY7503 on Damage-Induced HUVECs

To identify the bioactive substances in the HY7503 strain, we isolated SLPs and extracellular vesicles (exosomes) from HY7503 and examined their effects on HUVECs treated with TNF-α. The results of the MTT assay, conducted using HUVECs, revealed that the SLPs and exosomes were non-toxic at concentrations below 10 μg/mL and a dilution ratio of 1/10^4^, and the cell viability was approximately 82.61–94.93% (Figure 9A). TNF-α was used to inhibit NO production in HUVECs; NO production in HUVECs significantly decreased by 16.28 ± 0.01% compared to that in the control (*p* < 0.001; Figure 9B). The SLPs increased the NO levels in a dose-dependent manner (39.53–67.44%; *p* < 0.005; Figure 9B); the exosomes significantly increased NO production at the dilution ratio of 1/10^4^ (39.53 ± 6%; *p* < 0.005; Figure 9B). Subsequently, the expression levels of *eNOS*, *ICAM-1*, and *VACM-1* were analyzed using RT-PCR. Treatment of the cells with TNF-α led to a significant decrease in *eNOS* levels (*p* < 0.01; Figure 9C) and a significant increase in *ICAM-1* and *VCAM-1* levels compared to those in the control (*p* < 0.001; Figure 9D,E). Treatment of the cells with SLPs enhanced the *eNOS* expression levels in a dose-dependent manner (*p* < 0.05), whereas treatment of the cells with exosomes did not result in any significant changes (Figure 9C). Treatment of the cells with SLPs significantly reduced *ICAM-1* and *VCAM-1* expression levels in a dose-dependent manner (*p* < 0.05; Figure 9D,E). Treatment of the cells with exosomes resulted in a decrease in *ICAM-1* levels in a dose-dependent manner (*p* < 0.05; Figure 9D); however, the *VCAM-1* expression levels changed significantly only at a dilution ratio of 1/10^4^ (*p* < 0.01; Figure 9E).

## 3. Discussion

Endothelial dysfunction is a type of nonobstructive coronary artery disease (CAD) that causes chronic chest pain by contracting large blood vessels on the surface of the heart; the vasoconstriction is caused by a decrease in NO production in the blood vessel wall [19]. This endothelial dysfunction occurs in the early stages of atherosclerosis development; however, recent studies have shown that endothelial dysfunction is also associated with hypertension, inflammation, heart failure, and diabetes [5,20]. Therefore, studies on the alleviation of endothelial dysfunction are actively underway [21,22,23]; however, few studies have been conducted on the use of probiotics and their bioactive substances for the alleviation of endothelial dysfunction. In this study, we investigated the effects of HY7503 and its bioactive components on endothelial dysfunction. We examined the effects of HY7503 on the production of NO and the regulation of CAMs using in vitro (TNF-α-treated HUVECs) and in vivo (angiotensin II-injected mice) experiments.

The beneficial effects of LABs on human health have been demonstrated in various studies, and LABs are commonly used as functional foods [9]. To identify novel functional probiotics, we isolated five new LAB strains and determined their health benefits. Antioxidant and anti-inflammatory activities were examined using functional screening tests, and vascular endothelial cell activity was analyzed. The DPPH assay results showed that three LAB strains (*L. reuteri* #1, #2, and #3) exhibited a dose-dependent increase in antioxidant activity; among these strains, *L. reuteri* #1 had the highest antioxidant activity (Figure 1). In contrast, the other two strains (*L. paracasei* #4 and #5) did not show any differences in antioxidant activities at any concentration (Figure 1). Previous studies have shown that the antioxidant activities of different LABs vary significantly [24]. We compared the anti-inflammatory properties of the five LAB strains. Inflammation is primarily regulated by proinflammatory mediators as a defense mechanism against tissue damage caused by bacterial infections and chemical or physical damage [25]. LPSs are some of the most potent activators of macrophages and promote the production of NO and proinflammatory cytokines [26]. We observed a significant increase in the levels of NO produced by LPS-treated RAW264.7 cells (Figure 2B), which is consistent with the findings of a previous study [27]. All five LAB strains significantly reduced NO production (Figure 2B); this result confirmed the anti-inflammatory effects of the five LAB strains. Next, we determined the effects of the five LAB strains on vascular endothelial cell dysfunction caused by inflammation. In HUVECs treated with TNF-α, NO production was significantly reduced compared to that in the control group (Figure 3B). TNF-α increases ROS production via a proapoptotic pathway through TNF-α-induced signaling complex II, causing oxidative stress, thereby inhibiting NO production in vascular endothelial cells [28]. Two LAB strains (*L. reuteri* #1 and #2) promoted NO production, and *L. reuteri* #1 showed the highest NO production levels (Figure 3B). NO is produced by eNOS in endothelial cells and reduces blood pressure and has antithrombotic effects [29,30]. Therefore, the promotion of NO production by LAB strains can positively affect endothelial function. Collectively, these results indicate that *L. reuteri* #1 has excellent antioxidant and anti-inflammatory properties, which may help in the alleviation of endothelial dysfunction.

To confirm the efficacy of *L. reuteri* HY7503, which was selected on the basis of the in vitro study results, we performed in vivo experiments using mice. Mice were intraperitoneally injected with Ang II for four weeks to induce endothelial dysfunction. Ang II is a key factor regulating vascular endothelial dysfunction [31]. Levels of Ang II in the blood increase during hypertension and diabetes, which affects the thickness of the vascular media [32]. After one week, the Ang II-treated mice showed a decrease in weight compared with that observed in the CON group (Figure 4A). This weight loss commonly occurs after the administration of Ang II [33,34]. The aortic media thickness and MT/LD ratio in the Ang II-treated group were significantly higher than those in the CON group (Figure 4C,D). Blood vessel thickness is considered the primary indicator of vascular endothelial dysfunction. Factors such as hypertension, diabetes, and inflammation can thicken the blood vessel media; these structural changes are fatal in patients with cardiovascular diseases such as atherosclerosis [35,36]. Also, Ang II has been found to induce superoxide and ROS production and reduce NO bioavailability through the activation of NADPH oxidase. This, in turn, causes vascular structural changes such as vascular hypertrophy and fibrosis [37,38]. The oral administration of *L. reuteri* HY7503 did not affect Ang II-induced weight loss but reduced the increase in aortic media thickness and MT/LD caused by Ang II in a dose-dependent manner (Figure 4C). Additionally, treatment with *L. reuteri* HY7503 significantly increased the levels of serum NO (Figure 5A) and eNOS, the main enzyme that produces NO [39].

In the blood, NO is an important endogenous vasodilator that antagonizes vascular disorders caused by Ang II [40]. NO is a signaling factor produced by eNOS and performs various functions, such as blood flow and circulation control, thrombosis, and inflammation [39]. Endothelial dysfunction interferes with NO production by eNOS and reduces the bioavailability of NO [41]. Furthermore, the intake of *L. reuteri* HY7503 significantly reduced the gene expression levels of *ICAM-1* and *VCAM-1* in the aorta and the levels of TNF-α and IL-6 in the serum. Cell adhesion molecules (CAMs), such as ICAM-1 and VCAM-1, are produced by inflammatory cytokines; their levels indicate the inflammatory process occurring in the vascular endothelium; these are also considered early indicators of atherosclerosis [42]. Their levels increase upon Ang II treatment, and these are known to play a key role in endothelial dysfunction [43]. TNF-α and IL-6 are inflammatory cytokines that promote the expression of CAMs and inhibit NO-mediated endothelium-dependent vascular relaxation [44]. Inflammatory cytokines are the main cause of endothelial dysfunction, affecting all blood vessels. In particular, proinflammatory cytokines, such as TNF-α and interleukin-6 (IL-6), directly affect the expression levels of ICAM-1, VCAM-1, and eNOS and inhibit the production of NO in endothelial cells [44,45]. The inflammatory cytokine (TNF-α and IL-6) levels decreased upon the ingestion of *L. reuteri* HY7503, which was associated with a decrease in the gene expression levels of *ICAM-1* and *VCAM-1*; moreover, decreased inflammatory cytokines are also associated with decreased NO levels in the serum.

Subsequently, we identified the mechanism underlying the improvement in endothelial dysfunction by *L. reuteri* HY7503 in vitro. In HUVECs treated with TNF-α, *L. reuteri* HY7503 significantly increased eNOS-related gene expression levels (Figure 8A) and reduced the gene expression levels of *ICAM-1* and *VCAM-1* (Figure 8B,C). This result was consistent with the results of animal experiments; this confirmed that the mechanism by which *L. reuteri* HY7503 alleviates endothelial dysfunction involves an increase in NO production through an increase in *eNOS* expression levels and a decrease in *ICAM-1* and *VCAM-1* expression levels. Meanwhile, the efficacy of *L. reuteri* HY7503 in vitro was the highest at 1 ×10^6^ CFU/mL, possibly because this concentration was the highest concentration that could be effective. However, further studies are needed to determine the exact reason why the in vitro efficacy of *L. reuteri* HY7503 is not dose-dependent.

One of the most intriguing findings of this study was the identification of bioactive substances in *L. reuteri* HY7503. The isolation of SLPs and extracellular vesicles (exosomes) provided insights into the mechanisms underlying the biological activity of *L. reuteri* HY7503. SLPs are cell surface-associated proteins found in several Lactobacillus species and are known to play an important role in adherence to intestinal cells and host immunomodulation through the role of the adhesiveness of strain to cell [46,47]. Exosomes are 30 to 200 nm vesicular vesicles secreted by multiple-vesicular endosomes, which interact with target cells to influence various physiological mechanisms such as inflammation and immune response [48]. SLPs significantly enhanced NO production, upregulated *eNOS* expression, and reduced *ICAM-1* and *VCAM-1* expression levels in a dose-dependent manner (Figure 9B–E). In contrast, exosomes were less effective than SLPs in modulating these pathways; the notable effects of exosomes were observed only at high concentrations (Figure 9B–E). This suggests that SLPs might play an important role in the vascular protective effects of *L. reuteri* HY7503, although exosomes may contribute to its overall bioactivity in certain contexts. However, the pathways by which SLPs and exosomes directly affect the improvement in endothelial dysfunction are still limited. Therefore, further studies are needed to examine the absorption and molecular mechanisms of SLPs and exosomes in vascular endothelial dysfunction.

## 4. Materials and Methods

### 4.1. Isolation and Preparation of Lactic Acid Bacterial Strains

Five LAB strains were obtained from the probiotic strain library of Hy Co., Ltd.; all strains were isolated from raw milk in the Yeong-dong region of Korea. Sterile phosphate-buffered saline (PBS; 9 mL; Welgene, Gyeongsan, Republic of Korea) was added to 1 mL of raw milk and vortexed. The mixture was serially diluted with PBS, spread on De Man–Rogosa–Sharpe (MRS) agar plates (BD Difco, Sparks, MD, USA), and incubated for 48 h at 37 °C under anaerobic conditions. To obtain pure isolates, five cultured colonies were randomly selected and streaked onto fresh MRS agar plates. These LAB strains were maintained at −80 °C as frozen stocks in 20% (*v*/*v*) glycerol. For the experiments, five strains were harvested by incubating MRS broth medium (BD Difco) at 37 °C for 24 h and then centrifuging at 4000× *g* for 20 min. The harvested cell pellets were washed, resuspended in PBS, and used for in vitro assays. Mass-cultured LAB strains were used for in vivo studies. The LAB strains were used to count the number of viable cells after lyophilization, and 1 × 10^8^ or 1 × 10^9^ CFU/kg/day of the LABs were administered to mice.

### 4.2. 2,2-Diphenyl-1-Picrylhydrazyl (DPPH) Radical Scavenging Activity

DPPH radical scavenging was analyzed with slight modifications (concentration of DPPH solution), as described in a previous study [49]. Five strains were prepared at different concentrations (10^6^–10^8^ CFU/mL) and ascorbic acid was used as the standard. The DPPH solution (0.36 mM) was mixed with the samples (1:1; *v*/*v*) and allowed to react at room temperature for 45 min. The absorbance of each mixture was measured at 517 nm using a BioTek^®^ Synergy HTX multimode reader (Agilent Technologies, Santa Clara, CA, USA). The antioxidant activity was calculated using the following formula:DPPH radical scavenging activity (%) = (Ac − As)/Ac × 100

As: absorbance of the sample; Ac: absorbance of the control.

### 4.3. Cell Culture

HepG2 cells were obtained from the Korean Cell Line Bank (KCLB, Seoul, Republic of Korea), and HUVECs were obtained from the American Type Culture Collection (ATCC, Manassas, VA, USA). HepG2 cells were grown in Dulbecco’s modified Eagle’s medium (DMEM) with high glucose containing 10% fetal bovine serum (Gibco, Grand Island, NY, USA) and 1% penicillin–streptomycin (Gibco, Grand Island, NY, USA). The HUVECs were grown in F-12K medium (Catalog No., ATCC 30-2004; ATCC) containing 10% fetal bovine serum (Gibco), 10 mg/mL of 1% stock heparin solution (Sigma-Aldrich, St. Louis, MO, USA), and 30 mg/mL of 0.1% endothelial cell growth supplement (ECGS; Corning, NY, USA). These cells were grown at 37 °C under 5% CO_2_ in completely humidified air, and the medium was changed every two days; the cells were subcultured at 80–90% confluence.

### 4.4. Cell Viability and Nitric Oxide (NO) Production

The cytotoxicity of HepG2 cells and HUVECs treated with the five LAB strains was assessed by the MTT assay with slight modifications, as described in a previous study [50]. The cells were incubated in 96-well plates with the growth medium until they reached 80% confluence. The cells were then cultured in 200 µL of serum-free DMEM or F-12K medium for 24 h without any sample (control) or with samples (5 LAB strains; 1 × 10^5^–10^8^ CFU/mL). After incubation, 23 µL of the MTT stock (0.5 mg/mL in PBS) was added to each well, and the mixtures were allowed to stand for 4 h. The supernatant was removed after centrifugation at 1500 rpm for 5 min, and 150 μL of DMSO was added to each well to dissolve the formazan. The absorbance was measured at 540 nm using the BioTek^®^ Synergy HTX multimode reader (Agilent). NO production was expressed as a percentage of the production observed in the control.

The NO produced by RAW264.7 cells and HUVECs was detected by measuring the NO_2_^−^ concentration, an indicator of NO synthesis; the Griess reagent assay was used for the analysis, with slight modifications made to the protocol mentioned in the previous study [51]. The cells were incubated in 96-well plates with the growth medium until they reached 80% confluence. The cells were cultured in 200 μL of serum-free DMEM or F-12K medium for 1 h without any sample (control) or with samples (five LAB strains; 1 × 10^7^ CFU/mL). Subsequently, 23 μL of LPS stock (10 μg/mL) or TNF-α (100 mg/mL) was added to each well except the control. After 23 h of incubation, the culture supernatant was obtained, mixed with Griess reagent (Sigma-Aldrich), and incubated with shaking at room temperature for 15 min. The absorbance was measured at 540 nm using a BioTek^®^ Synergy HTX multimode reader (Agilent).

### 4.5. Animal Experiments

Eight-week-old (*n* = 24) male C57BL/6 J mice were purchased from DooYeol Biotech (Seoul, Republic of Korea). The mice were housed individually in a room with a controlled environment (temperature, 20–22 °C; humidity, 40–60%; 12 h light/dark cycle) and consumed a rodent diet (crude protein, 18.4%; fat, 6%; carbohydrates, 44.2%; crude fiber, 3.8%; neutral detergent fiber, 14.7%; and ash, 5.5%; Envigo, Indianapolis, IN, USA) and tap water ad libitum for one week during the acclimation period. After this period, the mice were divided into four groups (*n* = 6): control mice (CON), mice injected with angiotensin II (Ang II), mice injected with angiotensin II and fed 1 × 10^8^ CFU of *L. reuteri* HY7503/kg/day (HY7503-L), and mice injected with angiotensin II and fed 1 × 10^9^ CFU of *L. reuteri* HY7503/kg/day (HY7503-H). The CON and Ang II groups were orally administered saline using disposable zondes; the HY7503-L and HY7503-H groups were orally administered each sample once daily for five weeks using disposable zondes. After one week of oral administration, angiotensin II was injected intraperitoneally (IP) once a day into all groups except the CON group. The body weight and food intake were measured weekly. On the last day of the experiment, mice were sacrificed using carbon dioxide (CO_2_) The animal study was conducted according to the guidelines of Hy Co., Ltd. (Yongin-si, Republic of Korea) and approved by the Institutional Animal Care and Use Committee of Hy Co., Ltd. (IACUC approval number: AEC-2024-0006-Y; 14 June 2024).

### 4.6. Histological Analysis

Aortic tissue was collected at the end of the study, fixed in 10% (*v*/*v*) formalin solution, and embedded in paraffin. Sections were cut and mounted on slides before staining with hematoxylin and eosin (H&E). The thickness of the aortic media and LD in each mouse were measured, and three representative images were captured at 40× magnification. The thickness of the aortic media and LD were determined by calculating the mean of 100 values per group using a Motic digital microscope image analysis system (Motic Optical Instruments Co., Ltd., Xiamen, China).

### 4.7. Determination of NO and Cykokine Levels in Serum

Whole-blood samples were collected via cardiac puncture, and serum was separated via centrifugation at 2000× *g* for 15 min at 4 °C. The serum was then stored at −80 °C until further use. Serum NO levels were measured as previously described in the in vitro cell study. The obtained serum was mixed with the Griess reagent (Sigma-Aldrich) and incubated with shaking at room temperature for 15 min. The absorbance was measured at 540 nm using the BioTek^®^ Synergy HTX multimode reader (Agilent). NO production is expressed as a percentage of that in the CON group.

Serum cytokine levels were measured using an ELISA kit (BD Biosciences, San Jose, CA, USA) according to the manufacturer’s instructions. Serum was used at a 100-fold dilution for analysis. The capture antibodies were diluted with the coating buffer, added into 96-well plates (100 μL), and coated at 4 °C for 24 h. The next day, the antibodies were washed three times using a washing buffer, the assay diluent (200 μL) was added, and the solution was kept at room temperature for 1 h. After three repetitions of washing, the sample or standard solution was added (100 μL), and the mixture was incubated at room temperature for 2 h. After five washes, the working detector was added, and the cells were incubated at room temperature for 1 h. Finally, after seven washes, the substrate solution was added and incubated in the dark for 30 min, and then, the stop solution was added; the absorbance was analyzed within 30 min of adding the stop solution. The absorbance was measured at 450 nm using the BioTek^®^ Synergy HTX multimode reader (Agilent).

### 4.8. RNA Extraction, cDNA Synthesis, and Real-Time Polymerase Chain Reaction (RT-PCR)

Total RNA from aorta tissue (20 mg) was isolated using the Easy-spin Total RNA Extraction Kit (iNtRON Biotechnology, Gyeonggi, Republic of Korea), according to the manufacturer’s instructions. We used 2 μg of total RNA to synthesize the cDNA template using an Omniscript RT Kit (QIAGEN, Hilden, Germany). The resulting cDNA template was used for RT-PCR.

Gene expression analysis was performed using the QuantStudio 6 RT-PCR program (Thermo Fisher Scientific, Waltham, MA, USA). Quantifications of endothelial NOS (*NOS3*; Mm00435217_m1), intercellular adhesion molecule-1 (*ICAM-1*; Mm00516023_m1), and vascular cell adhesion molecule-1 (*VCAM-1*; Mm01320970_m1) were performed using gene-specific primers purchased from Applied Biosystems (Middlesex County, MA, USA). Expression data were normalized to glyceraldehyde 3-phosphate dehydrogenase (GAPDH) expression levels using the comparative *C*_T_ method.

### 4.9. Isolation of Surface-Layer Proteins (SLPs) and Extracellular Vesicles (Exosomes)

For the preparation of SLPs, *L. reuteri* HY7503 was grown in MRS broth for 24 h, followed by centrifugation at 10,000× *g* for 10 min at 4 °C; the bacterial pellets were collected. The pellet was washed twice with PBS, and a LiCl (5 M; Sigma-Aldrich) solution was added to the pellet and incubated with shaking at 200 rpm for 1 h. After incubation, the supernatant was obtained by centrifugation at 16,000× *g* for 30 min at 4 °C and vacuum filtrated (0.45 μm). The filtrate was dialyzed for 24 h in PBS using SpectraPor^®^ dialysis membrane (molecular weight cutoff [MWCO]: 12–14 kDa; Thermo Fisher Scientific, Waltham, MA, USA). SLPs were obtained by lyophilization and then stored at −80 °C until further use.

Extracellular vesicles (exosomes) from *L. reuteri* HY7503 were isolated using a Total Exosome Isolation kit (Thermo Fisher Scientific, Waltham, MA, USA), according to the manufacturer’s instructions. *L. reuteri* HY7503 cultures were grown for 24 h in MRS broth and centrifuged at 2000× *g* for 30 min to remove cells and debris. The obtained cell-free culture medium was mixed well with the reagent (2:1; *v*/*v*) and then incubated at 4 °C overnight. After incubation, the samples were centrifuged at 10,000× *g* for 1 h at 4 °C. Exosomes were obtained from pellets by aspirating the supernatant. The pellet was resuspended in a convenient volume of PBS and then stored at −20 °C until further use. The resuspension ratio was divided from 1/10^3^ to 1/10^6^ based on the volume of the culture medium.

### 4.10. Statistical Analysis

All data are presented as means ± standard deviations (SDs). To determine the statistical significance of the difference between the results for the control and angiotensin-treated groups and between the angiotensin-treated and LAB-treated groups, unpaired Student’s t-tests was performed. All statistical analyses were performed using SPSS version 26.0 (IBM Corp., Somers, NY, USA).

## 5. Conclusions

In conclusion, the in vivo experiment results showed that the oral administration of *L. reuteri* HY7503 improves endothelial dysfunction by suppressing the increase in thickness of blood vessels; these effects are mediated via an increase in NO levels in serum and *eNOS* expression levels in aorta and the inhibition of the production of inflammatory cytokines (TNF-α and IL-6) and CAMs (*ICAM-1* and *VCAM-1*). In addition, the separation of bioactive substances, including SLPs and exosomes, from HY7503 revealed that SLPs played an important role in determining the efficacy of *L. reuteri* HY7503. Exosomes also had some beneficial effects; however, these effects were less pronounced than those of SLPs. These findings highlight the therapeutic potential of L. reuteri HY7503 as a functional probiotic for the prevention and management of cardiovascular diseases through its ability to modulate inflammation and endothelial function. Future research should focus on further elucidating the molecular mechanisms of HY7503’s bioactive substances and exploring its clinical applications in human populations. Moreover, whether *L. reuteri* HY7503 affects blood pressure, another factor influencing the increase in blood vessel thickness, remains unknown. Further studies are needed to examine the correlation between blood pressure and the alleviation of endothelial dysfunction caused by *L. reuteri* HY7503.

## Figures and Tables

**Figure 1 ijms-25-11326-f001:**
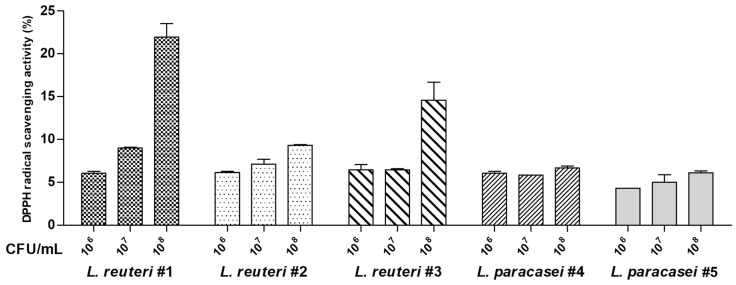
Screening of five lactic acid bacterial strains for their antioxidant activities. The 2,2-diphenyl-1-picrylhydrazyl (DPPH) radical scavenging activity of the five lactic acid bacterial strains. Five strains isolated from raw milk were listed as “strain name #number” in order. Data are expressed as means ± standard deviations (SD).

**Figure 2 ijms-25-11326-f002:**
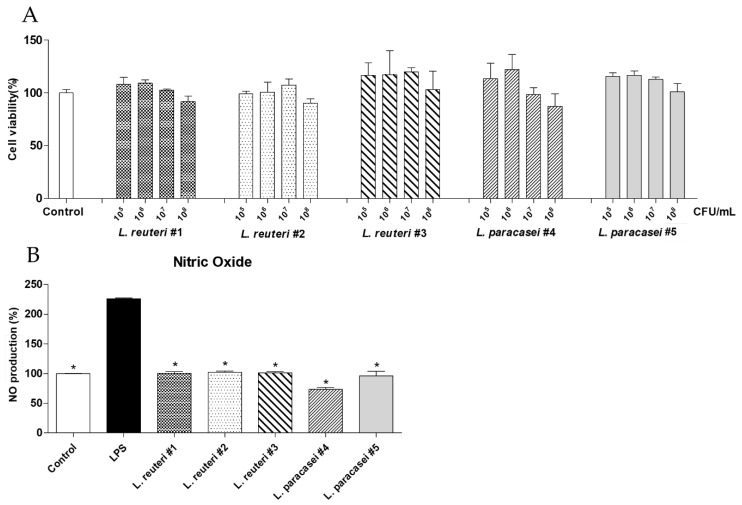
Screening of five lactic acid bacterial strains for anti-inflammatory activities. In RAW264.7 cells, (**A**) cell viability was assessed using the 3-(4,5-dimethylthiazol-2-yl)-2,5-diphenyltetrazolium bromide (MTT) assay, and (**B**) nitric oxide (NO) production in lipopolysaccharide (LPS)-treated cells was measured using the Griess reagent assay. Data are expressed as means ± SDs. Statistical analysis was performed using unpaired Student’s *t*-tests. Differences between results for the groups treated with the bacterial strains and the LPS group are indicated using *p*-values < 0.001 (*).

**Figure 3 ijms-25-11326-f003:**
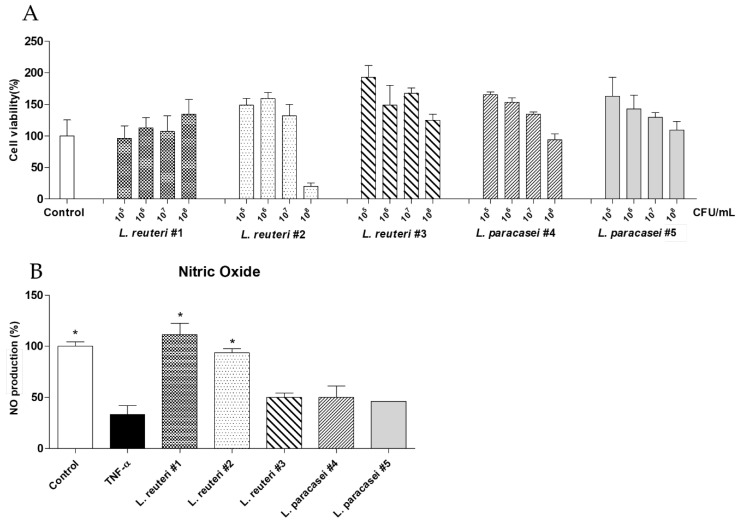
Screening of five lactic acid bacterial strains for vascular endothelial nitric oxide-promoting activity. In human umbilical vein endothelial cells (HUVECs), (**A**) cell viability was assessed using the MTT assay, and (**B**) NO production in tumor necrosis factor-alpha (TNF-α)-treated cells was measured using the Griess reagent assay. Data are expressed as means ± SDs. Statistical analysis was performed using unpaired Student’s *t*-tests. Differences between results for the groups treated with the bacterial strains and the TNF-α-treated group are indicated using *p*-values < 0.001 (*).

**Figure 4 ijms-25-11326-f004:**
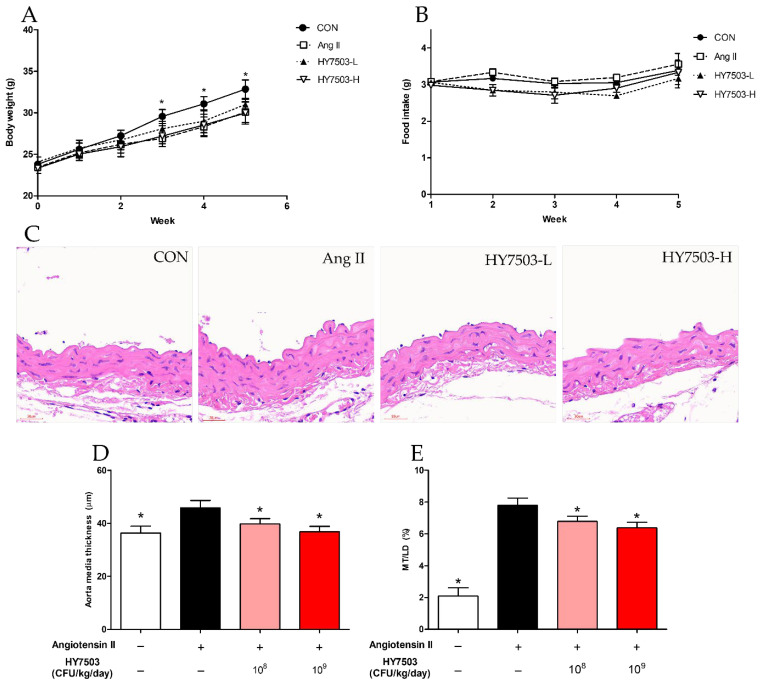
Effect of *L. reuteri* HY7503 on the aortic wall in mice with angiotensin II (Ang II)-induced endothelial dysfunction. (**A**) Body weight and (**B**) food intake were measured weekly for five weeks. (**C**) Representative images of stained aorta tissues. (**D**) The aorta media thickness (MT) and (**E**) ratio of MT to lumen diameter (LD) were measured using images captured with a digital microscope. Data are expressed as means ± SDs (*n* = 6 mice per group). Statistical analysis was performed using unpaired Student’s *t*-tests. (**A**) Differences between results for the Ang II-treated and other groups are indicated by *p*-values < 0.05 (*). (**D**,**E**) Differences between results for the Ang II-treated and other groups are indicated by *p*-values < 0.001 (*). Ang II, mice treated with angiotensin II; HY7503-L, mice treated with 10^8^ CFUs of *L. reuteri* HY7503/kg/day; HY7503-H, mice treated with 10^9^ CFUs of *L. reuteri* HY7503/kg/day.

**Figure 5 ijms-25-11326-f005:**
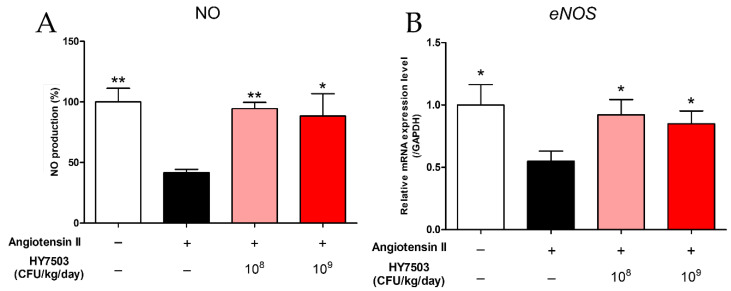
Effect of *L. reuteri* HY7503 on NO production in mice with angiotensin II-induced endothelial dysfunction. (**A**) Serum nitrite levels as an indicator of NO production were measured using the Griess reagent assay. (**B**) NO production-related gene expression (endothelial NO synthase, *eNOS*) in aorta tissue was analyzed using real-time polymerase chain reaction (RT-PCR). Data are expressed as means ± SDs (*n* = 6 mice per group). Statistical analysis was performed using unpaired Student’s *t*-tests. Differences between results for the Ang II-treated and other groups are indicated using *p*-values < 0.05 (*) and <0.001 (**).

**Figure 6 ijms-25-11326-f006:**
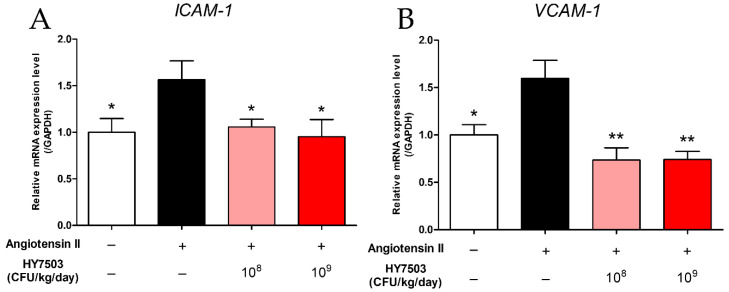
Effect of *L. reuteri* HY7503 on the expression of adhesion molecule-related genes in mice with angiotensin II-induced endothelial dysfunction. RT-PCR was performed to determine the mRNA expression levels of (**A**) intercellular adhesion molecule-1 (*ICAM-1*) and (**B**) vascular cell adhesion molecule-1 (*VCAM-1*). Data are expressed as means ± SDs (*n* = 6 mice per group). Statistical analysis was performed using unpaired Student’s *t*-tests. Differences between results for the Ang II-treated and other groups are indicated using *p*-values < 0.05 (*) and <0.005 (**).

**Figure 7 ijms-25-11326-f007:**
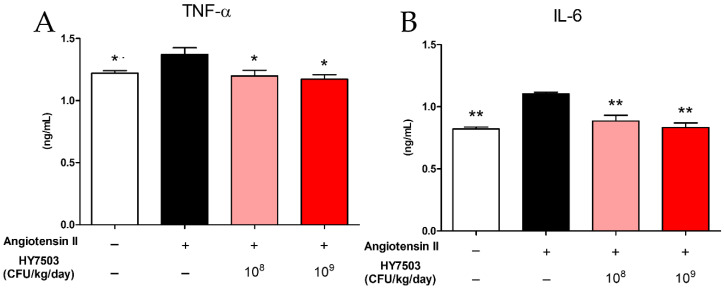
Effect of *L. reuteri* HY7503 on serum cytokine levels in mice with angiotensin II-induced endothelial dysfunction. (**A**) Tumor necrosis factor-α (TNF-α) and (**B**) interleukin-6 (IL-6) levels in the serum were measured using enzyme-linked immunosorbent assay (ELISA). Data are expressed as means ± SDs (*n* = 6 mice per group). Statistical analysis was performed using unpaired Student’s *t*-tests. Differences between results for the Ang II-treated and other groups are indicated using *p*-values < 0.05 (*) and <0.001 (**).

**Figure 8 ijms-25-11326-f008:**
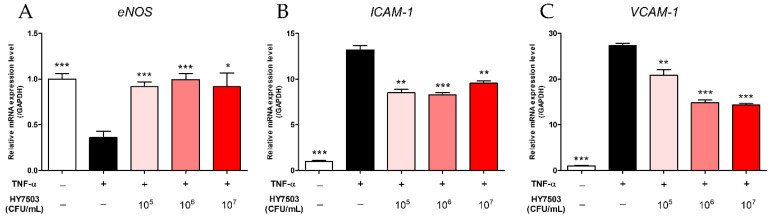
Effect of *L. reuteri* HY7503 on NO production and adhesion molecule-related gene expression in TNF-α-treated human umbilical vein endothelial cells (HUVECs). Real-time polymerase chain reaction (RT-PCR) was performed to determine the mRNA expression levels of (**A**) endothelial NOS (*eNOS*), (**B**) intercellular adhesion molecule-1 (*ICAM-1*), and (**C**) vascular cell adhesion molecule-1 (*VCAM-1*). Data are expressed as means ± SDs. Statistical analysis was performed using unpaired Student’s *t*-tests. Differences between results for the TNF-α-treated and other groups are indicated using *p*-values <0.05 (*), <0.01 (**), and <0.001 (***).

**Figure 9 ijms-25-11326-f009:**
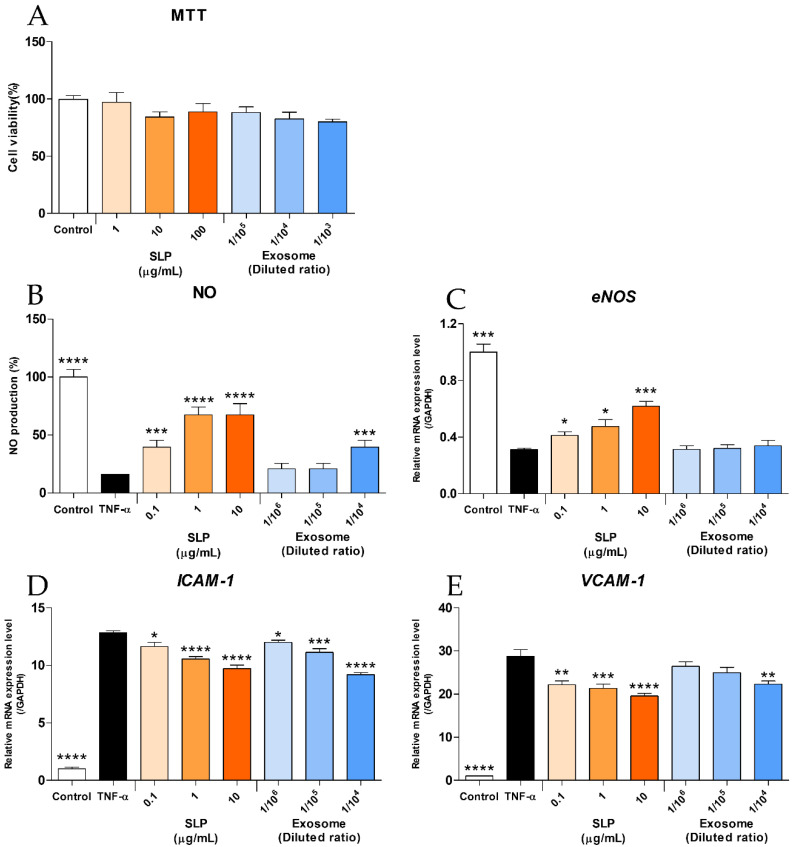
Effects of surface-layer proteins (SLPs) and extracellular vesicles (exosomes) of *L. reuteri* HY7503 on TNF-α-treated HUVECs. In HUVECs, (**A**) cell viability was assessed using the MTT assay, and (**B**) NO production in TNF-α-treated cells was analyzed using the Griess reagent assay. The expression levels of (**C**) endothelial NOS (*eNOS*), (**D**) intercellular adhesion molecule-1 (*ICAM-1*), and (**E**) vascular cell adhesion molecule-1 (*VCAM-1*) were determined using real-time polymerase chain reaction (RT-PCR). Data are expressed as means ± SDs. Statistical analysis was performed using unpaired Student’s *t*-tests. Differences between results for the TNF-α-treated and other groups are indicated using *p*-values <0.05 (*), <0.01 (**), <0.005 (***), and <0.001 (****).

## Data Availability

Data are contained within the article.

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
