# Peer review of "Limosilactobacillus reuteri HY7503 and Its Cellular Proteins Alleviate Endothelial Dysfunction by Increasing Nitric Oxide Production and Regulating Cell Adhesion Molecule Levels"

_ijms, 2024, doi:10.3390/ijms252011326_

Round 1

Reviewer 1 Report

Comments and Suggestions for Authors

This article makes a valuable contribution to the field of cardiovascular health by demonstrating the beneficial effects of Lactobacillus reuteri HY7503 on endothelial dysfunction, a key factor in the development of cardiovascular diseases such as atherosclerosis. The results suggest that this probiotic strain not only exhibits significant antioxidant and anti-inflammatory properties but also improves endothelial function by increasing nitric oxide (NO) production, a crucial vasodilator. Additionally, the study identifies key bioactive substances, such as surface-layer proteins (SLP), which play an important role in the observed vascular protective effects. Overall, the research highlights the therapeutic potential of L. reuteri HY7503 as a functional probiotic for the prevention and management of cardiovascular diseases.

INTRODUCTION:  

 Hypothesis Statement: Consider providing a more direct hypothesis or research question near the end of the introduction. This will give the reader a clearer understanding of what the study aims to demonstrate. the hypothesis could be more explicitly framed to highlight the novelty of this research. You mention that the mechanisms behind LAB effects on endothelial function remain unstudied, and this could be emphasized more clearly as the research gap your study aims to fill.

  • Grammar/Style: There are minor grammatical issues, such as the sentence “Probiotics, particularly lactic acid bacteria (LABs), have garnered significant attention for their potential health benefits in addition to improvement of gut health,” which could be rephrased for clarity (perhaps: "Probiotics, especially lactic acid bacteria (LABs), have attracted attention not only for improving gut health but also for their potential broader health benefits").
  • Referencing: Be sure that all references are up-to-date, especially since the field of microbiome and probiotics research is rapidly evolving. The use of citations from 2016-2018 seems appropriate, but consider whether there are more recent studies that can further support your claims.
  • The literature is adequately cited to support your claims. However, there are a few areas where elaboration could strengthen the review:
  • Probiotics and endothelial function: More emphasis could be placed on existing research showing direct or indirect links between probiotics and endothelial health. You mention that some studies have demonstrated improvements in endothelial dysfunction in hypertensive rats, but a few more specifics would add weight to the argument.
  • Bioactive components of LAB strains: The discussion on surface-layer proteins (SLP), exosomes, and exopolysaccharides is interesting but a bit brief. Given that these substances are key to your hypothesis, a more in-depth explanation or examples from recent studies would be helpful. For instance, discussing how SLP might interact with endothelial cells or oxidative stress mechanisms could enhance the reader’s understanding.
  •  
  • RESULTS AND DISCUSSION:
  • MTT Assay Results: You mention that the selected strains were non-toxic, but the range of cell viability (96.43%–192.86%) suggests an unexpectedly high upper limit. It would be useful to clarify this, as values exceeding 100% could raise questions. Is this due to enhanced cell proliferation? A short explanation or note would help avoid confusion.
  • When discussing the dose-dependent results (e.g., DPPH activity, vascular thickness), it would be beneficial to provide more details on the dose-response relationship. What are the implications of this dose dependence for potential therapeutic use? Including this information would make the findings more applicable to future research or clinical settings.
  • In several parts of the results, you state that the differences were "significant" without providing specific p-values. While you do report p-values in some sections (e.g., Figure 4), consistency is important. It would be beneficial to include the exact p-values or ranges in all instances where significance is mentioned.
  • Mechanism of Action of SLP and Exosomes: You have identified SLP as the primary bioactive substance responsible for enhancing NO production and downregulating CAM expression. Expanding on the molecular pathways through which SLP exerts these effects would provide more depth to your discussion. For example, does SLP directly interact with eNOS, or does it act via secondary signaling molecules? Further elaboration on these molecular interactions would add scientific rigor to your conclusions.
  • Exosomes: While you mention that exosomes had some beneficial effects, their role remains somewhat ambiguous. It would be helpful to discuss possible reasons for their lesser efficacy compared to SLP and suggest hypotheses for future studies. This will help outline future directions for understanding the combined bioactive effects of both substances.
  • Clinical Translation: You briefly mention future research in human populations but expanding on the potential for clinical application would make the findings more impactful. For instance, could L. reuteri HY7503 be developed into a dietary supplement or therapeutic agent? What are the key barriers to translating these results into clinical practice? Offering suggestions on potential clinical trials or formulation strategies would bridge the gap between laboratory findings and real-world application.
  • Further Research on Blood Pressure: You note that it remains unclear whether L. reuteri HY7503 affects blood pressure. Given the known role of NO in blood pressure regulation, this could be a significant area for future investigation. Consider elaborating on how future studies might explore this connection, perhaps through blood pressure monitoring in animal models or early-stage human trials.
  • Molecular Mechanisms: You suggest further studies to elucidate the molecular mechanisms of HY7503’s bioactive substances. Providing specific suggestions for experimental approaches (e.g., transcriptomics, proteomics) would enhance the robustness of your recommendations and show a clear path forward for future research.
  • Relevance of Endothelial Dysfunction: You have correctly emphasized the importance of endothelial dysfunction in the early stages of atherosclerosis and its links to hypertension, heart failure, and diabetes. However, providing a brief explanation of how these conditions collectively increase cardiovascular risk would underscore the broader health implications of your findings. For instance, reinforcing how endothelial dysfunction is a precursor to more severe cardiovascular issues such as stroke or myocardial infarction could further highlight the importance of therapeutic interventions like L. reuteri HY7503.
  • Comparison to Other Probiotics: It would be useful to mention any studies that have explored other probiotic strains for alleviating endothelial dysfunction, even if they are limited. Highlighting how L. reuteri HY7503 compares to other probiotics studied for similar purposes would strengthen the argument for its potential therapeutic application. Additionally, referencing studies that focus on related LAB strains and their bioactive components (e.g., SLP, exosomes) could position your findings more clearly within the context of ongoing probiotic research.

Reviewer 2 Report

Comments and Suggestions for Authors

In the article submitted for review, „Limosilactobacillus reuteri HY7503 and its cellular proteins alleviate endothelial dysfunction by increasing nitric oxide production and regulating cell adhesion molecule levels”, the authors analyze the potential of LAB strains isolated from raw milk for their protective effects on the vascular endothelium and identify the bioactive substances present in these LAB strains. The manuscript is clearly written and presented in a well-organized manner. The literature cited is mainly recent and relevant publications. The figures are clear. The experimental design is adequate to test the hypothesis. Data are interpreted appropriately and consistently throughout the manuscript. Conclusions are consistent with the evidence and arguments presented. I suggest a few corrections to the article:

1.      Line 424-425. Please shortly characterize the modifications mentioned.

2.      Line 425. Please check that the stated concentration (106-10 CFU/mL) is written correctly.

3.      Line 481 The authors write ‘...Ang II groups were orally administered saline; the HY7503-L and HY7503-H groups were orally administered each sample once daily for five weeks...’. Please describe the technique used to administer the samples to the rats by oral route (by feeding tube?).

4.      Line 487. The authors should provide the exact date of approval of the animal experiment.

5.      The authors should describe the procedure for killing the animals.

6.      Literature should be prepared according to the authors' guidelines.

7.  I suggest the authors present the mechanism of action of L. reuteri HY7503 on the scheme.

Reviewer 3 Report

Comments and Suggestions for Authors

I congratulate the authors for the work carried out and here are my suggestions for the manuscript improvement:

In the abstract, highlight your main results (values) and clearly state your study’s objectives. What can be done in future investigations?

Lines 31-33: References are missing.

The Results and Discussion sections are clear and well-presented. However, some parts you mentioned in the Results section should be moved to Discussion. The Results section is not supposed to discuss your results, only present them. So, these parts should be moved to the Discussion section.

In section 4.5 I would like you to include more details about the laboratory where the animal experiments took place and its conditions.

The Conclusions section should be provided separately and the main study limitations and strengths should be provided.

Round 2

Reviewer 2 Report

Comments and Suggestions for Authors

The authors have revised the manuscript and responded to all comments. I recommend the article for further proceedings.